# Screening of Active Ingredients from Wendan Decoction in Alleviating Palmitic Acid-Induced Endothelial Cell Injury

**DOI:** 10.3390/molecules28031328

**Published:** 2023-01-30

**Authors:** Nan Xu, Muhammad Ijaz, Haiyan Shi, Muhammad Shahbaz, Meichao Cai, Ping Wang, Xiuli Guo, Lei Ma

**Affiliations:** 1Department of Pharmacology, School of Pharmaceutical Science, Shandong University, Jinan 250012, China; 2Laboratory of Chinese Medicine Preparation, Shandong Academy of Chinese Medicine, Jinan 250014, China; 3Department of Clinical Pharmacy, The First Affiliated Hospital of Shandong First Medical University, Jinan 250014, China; 4Shandong Provincial Qianfoshan Hospital, Jinan 250014, China; 5Department of Radiology, Qilu Hospital Affiliated to Shandong University, Jinan 250012, China; 6School of Pharmacy, Shandong University of Traditional Chinese Medicine, Jinan 250355, China; 7State Key Laboratory of Precision Measurement Technology and Instruments, Tianjin University, Tianjin 300072, China

**Keywords:** Wendan decoction, active ingredients, screening, anti-vascular endothelial cell injury

## Abstract

(1) Objective: Traditional Chinese medicine (TCM) plays an important role in the treatment of numerous illnesses. As a classic Chinese medicine, Wendan Decoction (WDD) encompasses a marvelous impact on the remedy of hyperlipidemia. It is known that hyperlipidemia leads to cardiovascular injury, therefore anti-vascular endothelial cell injury (AVECI) may be an underlying molecular mechanism of WDD in the cure of hyperlipidemia. However, there is no relevant research on the effect of WDD on vascular endothelial cells and its pharmacodynamic substances. Therefore, the purpose of this study was to investigate the protective effect of WDD on vascular endothelial cells. (2) Methods: The chemical constituents of WDD were determined by LC-MS/MS technology. The protective effects of 16 batches of WDD on samples from human umbilical vein endothelial cells (HUVECs) were evaluated. Finally, gray relation analysis (GRA) and partial least squares regression (PLSR) were used to analyze the potential correlation between chemical ingredients and AVECI. (3) Results: The results indicated that WDD had apparent protective effect on endothelial cells, and pharmacological properties in 16 batches of WDD tests were apparently discrepant. The GRA and PLSR showed that trigonelline, liquiritin, hesperidin, hesperetin, scopoletin, morin, quercetin, isoliquiritigenin, liquiritigenin and formononetin may be the active ingredients of AVECI in WDD. (4) Conclusions: WDD has a protective effect on endothelial cell injury induced by palmitic acid, which may be related to its component content. This method was suitable for the search of active components in classical TCM.

## 1. Introduction

In recent years, significant changes have taken place in diet and lifestyle. The incidence of metabolic diseases, including hyperlipidemia, has increased, and patients have high levels of triglyceride, phospholipid and total cholesterol [1]. Hyperlipidemia can significantly increase the risk of other metabolic diseases and reduce the quality of life. Studies have shown that the level of lipid peroxidation in patients with hyperlipidemia has increased, and the mechanism of oxidative stress was involved in abnormal lipid metabolism [2]. A large number of clinical and basic research data at home and abroad have proved that hyperlipidemia, especially hypercholesterolemia is one of the important risk factors of atherosclerosis and cardio-cerebrovascular diseases. It is a process of chronic pathological changes, mediated by a variety of factors and links, in which there is always an interaction between blood and blood vessel wall. Therefore, the study of the effect of dyslipidemia on blood cells and vascular endothelial cells and their interaction has important theoretical and application value for understanding the mechanism of the occurrence and development of atherosclerosis and putting forward effective counter measures.

HUVECs are monolayer flat cells, lining the inner surface of the vascular system, and are important components of vascular intima. They can synthesize, secrete and express a variety of antithrombotic and thrombogenic substances. Endothelium injury is regarded as a reason of pathogenesis and clinical sign of hyperlipidemic vascular injury, and a large number of essential and auxiliary metabolic disarrangements have been connected to endothelial injury [3,4,5,6]. Experimental and clinical research has indisputably associated the endothelium with the pathogenesis of hyperlipidemia. Therefore, the clinical evaluation of endothelial cells may be considered to evaluate vascular wellbeing (or illness) [7].

TCM has played a basic role in treatment of various ailments since its long-standing history of clinical utilization [8]. In China, hyperlipidemia has become an important factor endangering public health, and shows a younger trend. However, long-term use of hypolipidemic chemical drugs has greater side effects [9]. Therefore, it is urgent to find lipid-lowering drugs with good safety and low toxicity. The TCM meets this demand. It is an important task to research and develop new lipid-lowering TCM preparations, which are of great significance and broad prospects. Through the study, it was found that the extracts of millet leaves can inhibit the apoptosis of vascular endothelial cells induced by hyperlipidemia in model rats by regulating the expression of apoptotic proteins Bax and Bcl-2. Bax and Bcl-2 regulate two key proteins of apoptosis, and they regulate the susceptibility of cells to apoptosis. The increased expression of Bax leads to cell death, while the effect of Bcl-2 is the reverse. The extract of millet leaves can significantly up-regulate the expression of Bcl-2 protein and down-regulate the expression of Bax protein, so as to protect against vascular endothelial oxidative damage caused by hyperlipidemia [10]. Paeonol regulated HUVEC development via downregulating miR-338-3p expression. Interestingly, miR-338-3p targeted TET2 and inhibited TET2 expression. MiR-338-3p modulated ox-LDL-treated VEC growth through suppressing TET2 expression. We demonstrated that paeonol attenuated the effect of ox-LDL on the development of mice HUVEC via modulating miR-338-3p/TET2 axis [11]. Vast evidence is now available that non-esterified FFA, usually typified by PA, abruptly triggers endothelial cell apoptosis in vitro1 [12,13]. HUVEC could not be maintained longer than 2 days even at low PA (or other FFA) concentrations that are compatible with their normal levels in human blood.

WDD dates from “Bei ji qian jin yao fang (Essential Recipes for Emergent Use Worth A Thousand Gold)” (written by Sun Simiao of the Tang dynasty), and was recorded in the Catalogue of Classical Prescriptions (first batch) issued by the state administration of traditional medicine of China in 2018 (NATCM, 2018). This was one of the primary 100 classic Chinese pharmaceutical medicines discharged and consisted of Pineilia ternata (Thunb.) Breit. (Ban-Xia, BX), Glycyrrhiza uralensis Fisch. (Gan-Cao, GC), Citrus reticulata Blanco (Chen-Pi, CP), Phyllostachys nigra (Lodd.) Munro var. henonis (Mitf.) Stapf ex Rendle (Zhu-Ru, ZR), Citrus aurantium L. (Zhi-Shi, ZS), Zingiber officinale Rose. (Sheng-Jiang, SJ). This prescription is the basic prescription of “expectorant”. Clinically, it is generally utilized within the treatment of diabetes, chronic kidney disease and hyperlipidemi [14,15,16]. The role of vascular complications in the development and progression of hyperlipidemia was well documented in the pulmonary vascular dysfunction, endothelial dysfunction and atherosclerosis [17,18]. Therefore, we infer that the efficacy of WDD is linked to the protection of endothelial cells. However, there is no research on the effect of WDD on endothelial cells, and the pharmacodynamic substances are not clear.

The identification method of chemical constituents in WDD refer to a previous study [19]. In all, a total of 20 ingredients related to hyperlipidemia were elected as quantitative chemical markers [20,21,22,23,24]. In this study, 16 batches of 20 chemical components of WDD were analyzed by ultra-performance liquid chromatography-triple quadrupole tandem mass spectrometry (UPLC-QQQ-MS/MS)), and the optimal concentration of WDD in the treatment of AVECI was selected by MTT method. Then the 10 active components of WDD for the treatment of AVECI were determined by two mathematical analysis modes (GRA and PLSR).

## 2. Result

### 2.1. Content Determination of Components in WDD

The single mass spectra of 20 compounds were studied by pouring the relevant standard solutions into the mass spectrometer in both positive and negative ion modes. The optimization results are shown in Table 1. The chemical structures of 20 chemical ingredients and IS were shown in Figure 1.

The contents of 20 compounds in 16 batches of WDD were determined by UPLC-QQQ-MS/MS within 10 min. The degree speed had major upgrades over the HPLC method, so ensuring that we may checkout still more cases in one day. Figure 2 shows the chromatographic behavior of 20 compounds and IS in WDD. The specific outcomes are shown in Table 2.

#### Method Validation

The 20 primitive courses of action were precisely debilitated with 50% (*v*/*v*) methanol-water. The slopes, intercepts and correlation coefficients (*r*^2^) were obtained by weighted (1/x) linear regression. The estimation ranges (S/N ≥ 10), lower limit of quantitation (LLOQ) and straight twists of diverse chemical fixings are tabulated in Table 3. The relative standard deviations (RSD) of precision, repeatability and stability were all less than 15%, and the single solution was stable at room temperature for 24 h and the recovery rate was within acceptable limits.

### 2.2. Cell Experiments

#### 2.2.1. Screening of PA Concentration by MTT

To determine the optimal inhibitory concentration of PA on HUVEC cells, the results were expressed as inhibition rate.
Inhibition rate (%) = (OD_sample_ − OD_blank_)/OD_blank_ × 100%(1)

OD means Optical Density. The tests were carried out in triplicate. The data are shown as the mean ± SD. The *p* value of < 0.05 was considered significant. The inhibition rate of PA extracted from HUVEC are shown in Figure 3. The inhibition rates were as follows: 0 µmol/L PA(Blank): 0%, 50 µmol/L PA: 5.50%, 100 µmol/L PA: 14.05%, 150 µmol/L PA: 29.50%, 200 µmol/L PA: 45.12%, 250 µmol/L PA: 54.10%, 300 µmol/L PA: 62.34%. The experimental results showed that cell proliferation was dose-dependent with PA concentration compared with the control group. Therefore, 200 μmol/L PA concentration was used for subsequent tests.

#### 2.2.2. Cell Protective Effect of WDD

Inhibition rates were utilized as the benchmark, the inhibitory impact of the ideal concentration of WDD on HUVEC damage was decided.
Inhibition rate (%) = (OD_sample_ − OD_model_)/OD_model_ × 100%(2)

The results showed that there were no major changes within the cell state of the control bunch. When the concentration of each bunch of WDD was 0.25 mg/mL, 0.5 mg/mL and 1 mg/mL, the cell condition had clear hyperplasia, and when the consistence was 2 mg/mL or over, it had inhibitory impact on the cell. Subsequently, 1 mg/mL concentration was chosen for ensuing tests. The results of filtrating HUVEC cells shielded by different consistencies of WDD by MTT strategy are shown in Table 4.

### 2.3. Statistical Analysis

The results of GRA and PLSR are shown in Figure 4 and Table 5. It can be seen from Figure 4 that synephrine, isoquercitrin, naringin, daidzein, naringenin, glycyrrhizic acid, nobiletin, tangeretin, obacunone, and glycyrrhetinic acid had a negative correlation with the in vitro efficacy results. The ingredients of trigonelline, liquiritin, hesperidin, hesperetin, scopoletin, morin, quercetin, isoliquiritigenin, liquiritigenin and formononetin had a positive relationship with the in vitro adequacy results. Observing the GRA results, it can be seen that these were the primary ten compounds with great advancing impact on pharmacodynamic parameters in vitro.

## 3. Discussion

Since the prehistoric age, human beings have used natural products, such as animals and marine organisms, in medicines to alleviate and treat diseases. On the basis of fossil records, the past records of human use of flora as medicines may be traced back at least 60,000 years. In the wake of the progress of theoretical background, therapeutic theory and related technologies, as well as comprehension of the life sciences, a limpid comprehension of the active ingredients of TCM has proved to be possible [25]. TCM includes a wide application prospect within the treatment of complicated infections such as cancer and diabetes due to its points of interest in combined direction component of “multi-components, multi-targets and multi-approaches” [26]. An assortment of tissues, organs or different targets take part within the entire preparation of complicated affliction and shaping a complicated network regulation mechanism [27]. It has complex components and multiple targets, which is in line with the characteristics of TCM [28].

The inquiry about the viable components of TCM as a rule begins with the efficient division of chemical components, and its organic movement or pharmacodynamic impact was screened to clarify its compelling components [29], this strategy is time consuming. In this research, we used UPLC-QQQ-MS/MS technology to study the content of WDD. Combined with MTT method and mathematical statistical analysis, we screened 10 ingredients of WDD to protect endothelial cells. The results showed that WDD was concentration dependent in protecting endothelial cells. However, when the concentration was 2 mg/mL, the protective ability decreased. This may be because when the consistencies change, the condition of molecules in the liquor will also change. This may bring about the formation of complex structures or changes within the pharmacokinetics of the drugs and thereby reducing binding of drug to targets [30,31].

The agreeable impacts of “multicomponent, multichannel and multi-target” makes it difficult to inquire about the exercises of its pharmacodynamic materials [32]. At the same time, this is also one of the advantages of TCM. The 17 miRNAs in HUVECs that were raised by formononetin; the impact was biggest within the circumstances of miR-375, which was situated in an elevation backward-looking intergenic region between the cryba2 and Ccdc108 [33]. The move to communicate with miR-375 in HUVECs can increment hyperplasia and diminish apoptosis. The truth that formononetin incites the miR-375/RASD1/ERα input circuit in HUVECs, which at that pathway point is activated, may clarify why the isoflavone energized hyperplasia and limited apoptosis in HUVECs [34]. In addition, formononetin regulates ERK1/2 and P38 MAPK pathways, participates in the overexpression of EGR-1 transcription factor, and promotes endothelial cell repair and wound healing. The hesperidin hindered the expression of ERK, P38 MAPK and PI3K/AKT in VEGF-induced HUVECs. Hesperidin also inhibited the growth of aortic ring microvessels in mice in vitro. In conclusion, hesperidin inhibits endothelial angiogenesis by inhibiting PI3K/AKT, ERK and P38 MAPK signaling pathways [35]. In vivo studies further showed that isoliquiritigenin can inhibit the growth and angiogenesis of breast cancer, and inhibit VEGF/VEGFR-2 signal transduction, and increase the rate of apoptosis with little toxicity [36]. Isoliquiritigenin inhibited the expression of VCAM-1, e-selectin and PECAM-1 proteins induced by pro-inflammatory cytokines at the transcriptional level and weakened the interaction between THP-1 monocytes and endothelial cells. In this study, the decrease of VCAM-1 suggested the effect of isoliquiritigenin on endothelial function [37].

It is becoming evident that low-dose liquiritigenin can clear endocellular ROS, control or repress ROS interceded flag pathway, and ensure an arrangement of cells such as neurons, HUVECs, macrophages, hippocampal neurons, osteoblasts and hepatocytes [38]. Low-dose liquiritigenin antagonized HUVECs by inhibiting ROS mediated signal pathway [39]. HMEC-1 cells were cultivated in Matrigel for 24 h to observe the tube-formation. They were suddenly vascularized beneath Matrigel condition. The quercetin overwhelmingly quelled cells, practicality, relocation, VEGF expression and encouraged apoptosis in HMEC-1 cells. Moreover, concealment of miR-216a was found in HMEC-1 cells after quercetin incitement, in the interim miR-216a over expression abrogated the capacities of quercetin in HMEC-1 cells. The quercetin also deactivated PI3K/AKT and JAK/STAT pathways through altering miR-216a [40].

In conclusion, some chemical ingredients can act on the same pathway and target to treat diseases, but they can also act on different pathway targets. The ensemble curative effect of TCM is the outcome of the comprehensive action of the compounds. The active constituents of WDD are special compounds with multi-component, multiple targets and comprehensive regulation. Compared to a single-target strategy, the focal points of multi-objective strategy are becoming self-evident. In order to find out the dynamic substances of WDD, it was vital to understand the pharmacodynamics of its potential dynamic compositions. Therefore, we deduced that these active constituents are part of the effective constituents of WDD. The technique is highly efficient, high-speed, and economical. The quality marker (Q-marker) is the standard to evaluate the quality of traditional Chinese medicine compound, which should not be based on the ingredients of a few traditional Chinese medicines but should reflect the situation of the whole compound [41]. These ingredients can be used as reference standards for quality evaluation of WDD.

### Limitations

The research still has some shortcomings, such as small selection of the components. It still cannot fully represent the efficacy of WDD.

## 4. Materials and Methods

### 4.1. Materials

The 16 batches, comprising of WDD counting of BX, GC, ZS, CP, SJ, ZR were included in this research. These 16 batches of Chinese homegrown drugs come from different pharmaceutical markets of China. All homegrown solutions were distinguished by Teacher Jin Guangqian, TCM recognizable proof master of Shandong Institute of TCM. All the herbs met the necessities of Ch. P (2020 Version) as shown in Table 6.

Hesperidin (wkq20030407), tangeretin (wkq20041611), naringin (wkq21020606), nobiletin (wkq18020111), isoliquiritigenin (wkq18033006), formononetin (wkq18022712), naringenin (wkq21022409), glycyrrhetinic acid (wkq16070701) and glycyrrhizic acid (wkq16032502) were purchased from Sichuan Weikeqi Biotechnology Co., LTD., China. Daidzein (PS000251), isoquercitrin (PS001042), hesperetin (PS000219), trigonelline (PS000427), morin (PS020344), scopoletin (PS010525), liquiritigenin (PS010083) and liquiritin (PS012028) were provided by Chengdu Pusi Biotechnology Co., LTD., Sichuan Province, China. Obacunone (111923-201102) and furosemide (internal standard, IS) (100544-201503) were obtained from National Institutes for Food and Drug Control. Quercetin (100081-200406) and synephrine (0727-200105) were obtained by China National Institute for the Control of Pharmaceutical and Biological Products. All the reference substances were HPLC ≥ 98%. Acetonitrile for UPLC-QQQ-MS/MS was acquired from Merck KGaA (Darmstadt, Germany).

HUVEC cell line was purchased from Wuxi Bohe Biomedical Technology Co., LTD. (Jiangsu, China). RPMI-1640 basal medium was obtained from Hyclone Laboratory Media (Logan, UT, USA). Pancreatin was obtained from Shanghai Beyotime Biotechnology Co., LTD, China. The detection kit of MTT was bought from Sigma Company, Saint Louis, MO, USA. Penicillin-streptomycin was purchased from biological sharp, from Shanghai, China. Dimethyl sulfoxide (DMSO) was obtained from Invitrogen (Carlsbad, CA, USA).

### 4.2. Content Determination of Components in WDD

The preparation of medicinal materials was based on previous studies [42]. It was prepared according to the compatibility of medicinal materials and previous method [43]. Briefly, 10.0 g of BX, 10.0 g of ZS, 10.0 g of ZR, 15.0 g of CP, 20.0 g of SJ, and 5.0 g of GC were included to 840 mL of immaculate water and drenched for 30 min. The blend was decocted on high flame until bubbling, and after that decocted on low flame for 2 h. The above operations were repeated to obtain WDD solution and then concentrated with a rotary evaporator (Temperature: 60 °C; Rotational speed: 40–80 rpm). After merging and filtering, the supernatant was collected for concentration by Vacuum Freezing and then dehydrated in a drying oven at −50 °C for 72 h. This produced a dry powder. It was kept in a desiccator before use, and other batches of WDD powder were prepared using the same method.

The freeze-dried powder of WDD extricate (0.5000 g) was precisely weighed and ultrasonically broken up in methanol (30 min, at 25 °C). The arrangements were blended and balanced to a volume of 5 mL and diluted to the proper consistence. The test arrangements were sifted through a 0.22 µm channel film and the filtrate was stored at 4 °C for later UPLC-QQQ-MS/MS examination.

#### 4.2.1. Preparation of Standard Solution

All standard solutions were weighed accurately and diluted in methanol to form a reserve solution, which was stored at 4 °C. All reserve fluids were diluted to the desired concentration and then mixed immediately prior to analysis.

#### 4.2.2. UPLC-QQQ-MS/MS Conditions

LC–MS/MS investigations were carried out employing a Thermo Scientific Vanquish LC system connected to a TSQ Quantum Ultra triple quadrupole mass spectrometer (Thermo Scientific, Orlando, FL, USA) equipped with a heated electrospray ionisation source (H-ESI). Xcalibur 4.4 Software (Thermo Scientific) was used for method setup, data processing and reporting. The mass spectrometer was operated with a H-ESI interface handled in both positive and negative ionization modes and was utilized for the multiple reaction monitoring. Instrument settings were: positive ion: 3600 v, negative ion: 3400 v, sheath gas 0.3 mL/min, aux gas 0.3 mL/min, ion transfer tube temp 345 °C, vaporizer temp 350 °C.

Chromatographic fractionation was accomplished on a Thermo Hypersil GOLD C18 column (1.9 µm, 100 × 2.1 mm) at 30 °C, and versatile stage comprised 0.1% formic corrosive and water for dissolvable A and acetonitrile for dissolvable B at a stream rate of 0.3 mL/min. The gradient elution conditions of mobile phase B are as follows: 0–0.5 min: 5%, 0.5–2 min: 5–8%, 2–2.1 min: 40%, 2.1–4 min: 40–50%, 4–6 min: 50–60%, 6–6.1 min: 60–70%, 6.1–8 min: 70–80%, 8–8.1 min: 80–5%, 8.1–10 min: 5%. The sample size for analysis was 3 μL.

#### 4.2.3. Methodology Validation

The standard curve was established in the method validation part with internal standard method. The precision, stability, repeatability and recovery were investigated. The precision was calculated by 6 consecutive injections of the same sample solution, and repeatability was evaluated by six samples from the same source. The sample solution was put into the sample bottle at room temperature and injected for analysis at different times (0, 4, 8, 12, 16, 24 h) after preparation to observe the sample stability at room temperature. The measured WDD content was precisely weighed and added equal to 100% of the measured reference substance in WDD to determine the recovery of the sample.

### 4.3. Cell Experiments

#### 4.3.1. Screening of Palmitic Acid (PA) Concentration by MTT

The HUVECs were stored in a RPMI-1640 basal medium containing 10% FBS and other essential development variables in a humidified incubator with the climate containing 5% CO_2_ at 37 °C. The PA was broken down in DMSO (200 mM) as a stock solution and after that diluted with a medium containing RPMI-1640 basal medium with 10% FBS for later use (A: 0, B: 50 μmol/L, C: 100 μmol/L, D: 150 μmol/L, E: 200 μmol/L, F: 250 μmol/L, G: 300 μmol/L). The control treatment: cells were cultured with RPMI-1640 basal medium with 10% FBS. The experimental group added different concentrations of PA with the climate containing 5% CO_2_ at 37 °C. All the treatments were performed in triplicate. According to the above grouping, 10 μL of MTT was included in each well with the air brake containing 5% CO_2_ at 37 °C for 3–4 h in a dark incubator. After the liquid absorption, 150 μL DMSO was added and shaken at the shaking table at room temperature for 10 min. Finally, the optical density (OD) value at the same time point was calibrated at 492 nm wavelength by enzyme labeling instrument, and the measured OD value was used for cell viability analysis.

#### 4.3.2. Cell Protective Effect of WDD

The 16 batches of WDD powder were absolutely weighed. The volume was settled to 1 mL by PBS, and the microscopic organisms were sifted by 0.22 μm channel layer. Later the cells were included, and each batch of concentrated arrangement was weakened to 5 concentrations of 0.25 mg/mL, 0.5 mg/mL, 1 mg/mL, 2 mg/mL, 4 mg/mL by RPMI-1640 basal medium with 10% FBS. They were isolated into control group and the WDD treatment group. The cells treatment strategy was rehashed. All groups were in triplicate.

### 4.4. Statistical Analysis

#### 4.4.1. GRA

GRA’s significant concept is to select whether the filiation is close by characterising the geometric resemblance between the reference data course of action and a couple of comparison data columns. It mirrors the relationship degree between bends [44]. The data reflecting the pharmacodynamic behavior of WDD system were taken as the reference series, and the quantitative data of each component that might affect the pharmacodynamic behavior of WDD system were taken as the comparison series. GRA program (Grey Modeling_V3.0) was utilized to analyze, and the related coefficients between different components and in vitro pharmacodynamics evaluation records were obtained, which were sorted according to GRA the calculation condition.

#### 4.4.2. PLSR

PLSR could be a measurement strategy which brings the calculated variable amount and watched variable into an unused room to look out for a direct relapse demonstrate [45]. To encourage the unequivocal reliance between constituents and healing impact of WDD, the substance of 20 fundamental compositions in this study were contention and ampleness data were subordinate components. PLSR was obtained to analyze the information of 1 mg/mL and the regression coefficient.

## 5. Conclusions

A fast, sensitive and dependable LC-/MSMS strategy, combined with MTT and scientific factual examination strategy, was set up for the screening of numerous dynamic fixings of WDD with AVECI action. The results show that the extracts of WDD significantly increased the proliferation of endothelial cells and protected endothelial cells. However, the interaction between them can be additive, synergistic and antagonistic, and the mechanism remains unclear. More appropriate models should be included in the follow-up to conduct a comprehensive pharmacodynamics evaluation of WDD, laying a foundation for accurately and comprehensively revealing the active component group of WDD, so as to provide the basis for the rational use of WDD.

## Figures and Tables

**Figure 1 molecules-28-01328-f001:**
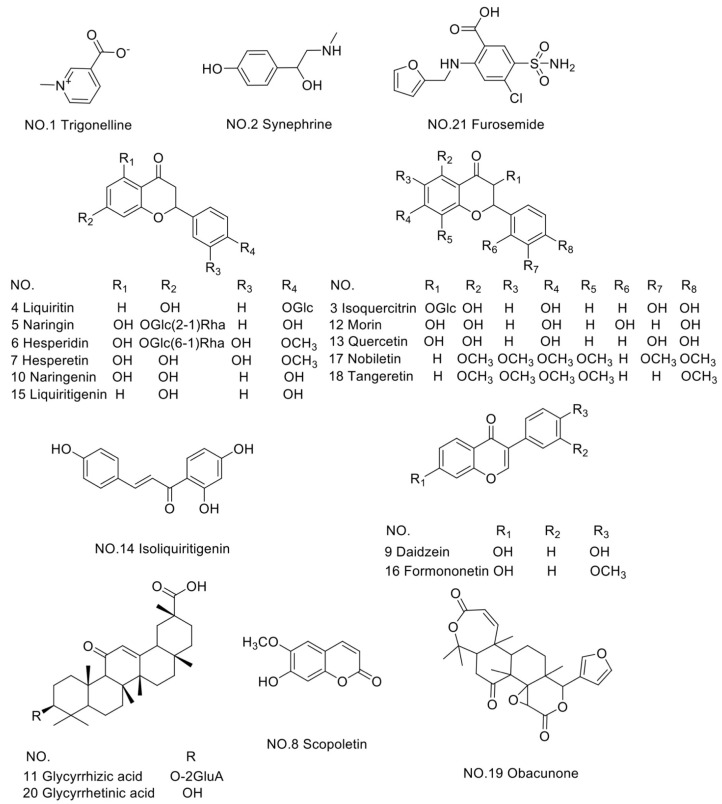
Structures of 20 chemicals analyzed in WDD and IS.

**Figure 2 molecules-28-01328-f002:**
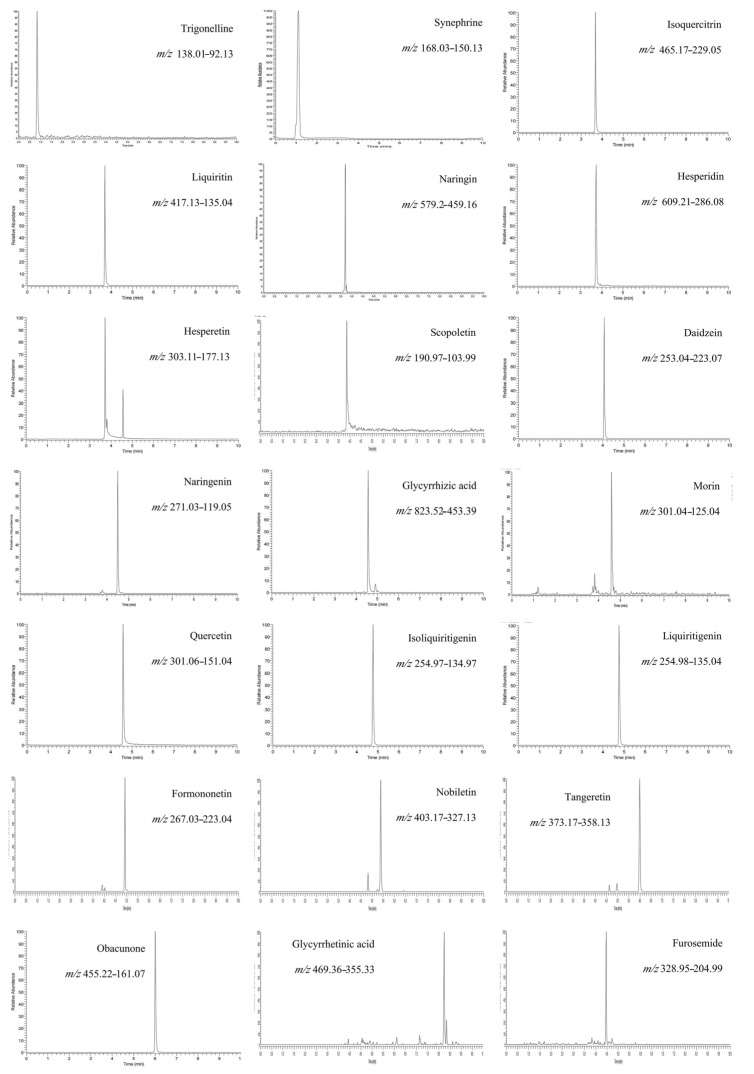
Chromatographic behavior of 20 compounds and IS.

**Figure 3 molecules-28-01328-f003:**
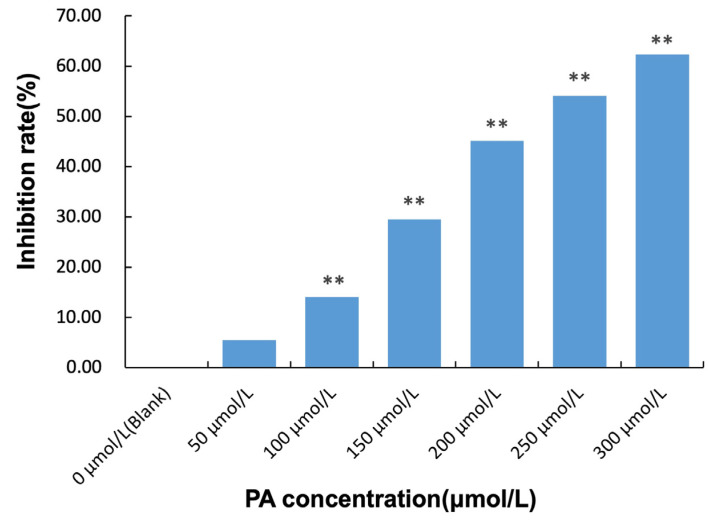
The inhibition rate of PA extracted from HUVEC. **: Compared with blank group, *p* < 0.01.

**Figure 4 molecules-28-01328-f004:**
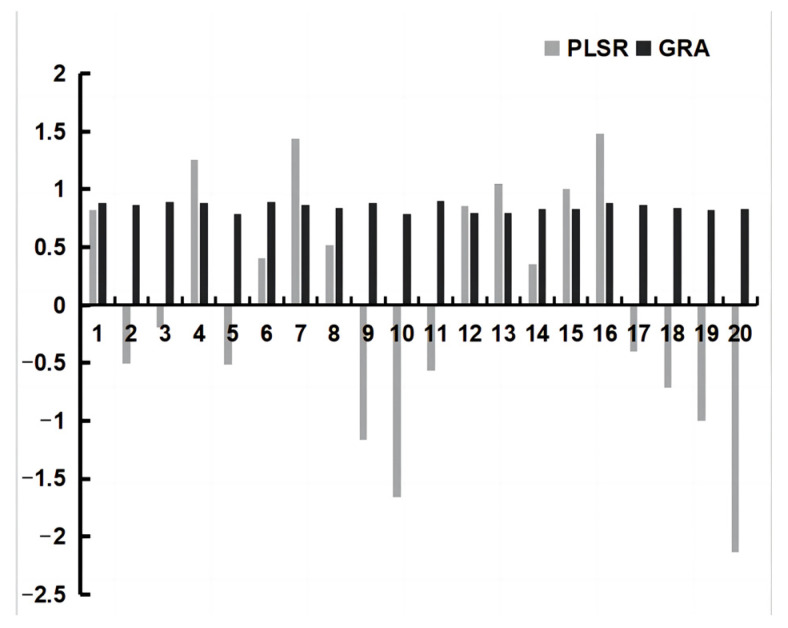
GRA and PLSR results for WDD compounds. GRA = Gray relation analysis; PLSR = Partial least squares regression; WDD = Wendan Decoction.

**Table 1 molecules-28-01328-t001:** List of quantified 20 compounds of WDD and IS parameters.

NO.	Compound	Formula	Mass	t_r_ (min)	Polarity	Precursor Ion (*m*/*z*)	Product Ion (*m*/*z*)	Collision Energy (V)	Radio Frequency Lens (V)
1	Trigonelline	C_7_H_7_NO_2_	137.14	0.84	_+_	138.01	92.13	19.91	97
2	Synephrine	C_9_H_13_NO_2_	167.21	0.97	_+_	168.03	150.13	8.16	49
3	Isoquercitrin	C_21_H_20_O_12_	464.38	3.69	_+_	465.17	229.05	44.22	106
4	Liquiritin	C_21_H_22_O_9_	418.39	3.69	_-_	417.13	135.04	31.41	147
5	Naringin	C_27_H_32_O_14_	580.53	3.71	_-_	579.2	459.16	22.99	234
6	Hesperidin	C_28_H_34_O_15_	610.56	3.73	_-_	609.21	286.08	41.73	170
7	Hesperetin	C_16_H_14_O_6_	302.28	3.74	_+_	303.11	177.13	16.84	133
8	Scopoletin	C_10_H_8_O_4_	192.17	3.89	_-_	190.97	103.99	25.05	83
9	Daidzein	C_15_H_10_O_4_	254.24	4.08	_-_	253.04	223.07	33.81	151
10	Naringenin	C_15_H_12_O_5_	272.25	4.51	_-_	271.03	119.05	26.52	124
11	Glycyrrhizic acid	C_42_H_62_O_16_	822.93	4.55	_+_	823.52	453.39	19.87	171
12	Morin	C_15_H_10_O_7_	302.24	4.59	_-_	301.04	125.04	20.25	137
13	Quercetin	C_15_H_10_O_7_	302.24	4.61	_-_	301.06	151.04	21.09	149
14	Isoliquiritigenin	C_15_H_12_O_4_	256.25	4.78	_-_	254.97	134.97	15.7	104
15	Liquiritigenin	C_15_H_12_O_4_	256.25	4.79	_-_	254.98	135.04	15.24	118
16	Formononetin	C_16_H_12_O_4_	268.26	4.96	_-_	267.03	223.04	32.21	127
17	Nobiletin	C_21_H_22_O_8_	402.4	5.43	_+_	403.17	327.13	32.17	182
18	Tangeretin	C_20_H_20_O_7_	372.37	6.03	_+_	373.17	358.13	19.07	167
19	Obacunone	C_26_H_30_O_7_	454.51	6.05	_+_	455.22	161.07	23.28	137
20	Glycyrrhetinic acid	C_30_H_46_O_4_	470.68	8.29	_-_	469.36	355.33	46.45	261
21	Furosemide	C_12_H_11_ClN_2_O_5_S	330.74	4.49	_-_	328.95	204.99	21.51	113

WDD = Wendan decoction; IS = Internal standard.

**Table 2 molecules-28-01328-t002:** The mean of the 20 chemical contents of WDD (*n* = 3, μg/g).

NO.	Compound	S1	S2	S3	S4	S5	S6	S7	S8	S9	S10	S11	S12	S13	S14	S15	S16
1	Trigonelline	1081.13 ± 115.55	1214.56 ± 98.98	968.28 ± 78.62	986.33 ± 54.57	2132.75 ± 248.64	1667.38 ± 253.09	2644.48 ± 250.17	1542.58 ± 182.98	2016.93 ± 213.55	2241.31 ± 238.07	1969.98 ± 67.36	798.42 ± 34.41	567.43 ± 71.72	1778.26 ± 45.44	2003.50 ± 273.90	1674.62 ± 261.21
2	Synephrine	13147.38 ± 1846.14	17359.46 ± 1855.39	18033.69 ± 165.49	12952.94 ± 1978.05	38277.40 ± 4178.07	36066.95 ± 4630.68	68086.80 ± 9517.01	18511.95 ± 2834.50	25981.23 ± 3620.37	58408.02 ± 3153.10	36739.46 ± 2112.75	27157.50 ± 965.88	6773.77 ± 960.51	37962.47 ± 1573.09	28646.49 ± 2927.31	38754.01 ± 4227.24
3	Isoquercitrin	123.85 ± 6.77	107.03 ± 2.40	123.63 ± 4.37	33.89 ± 3.42	80.13 ± 8.19	58.89 ± 6.21	95.03 ± 11.35	94.24 ± 2.64	100.42 ± 13.38	78.47 ± 13.19	96.81 ± 8.38	78.29 ± 6.24	91.47 ± 12.04	84.97 ± 4.08	60.87 ± 5.10	94.16 ± 10.12
4	Liquiritin	489.62 ± 58.75	1729.21 ± 240.56	601.38 ± 15.65	440.53 ± 33.29	995.53 ± 51.83	624.55 ± 59.79	1754.29 ± 222.48	764.55 ± 94.22	1717.53 ± 206.92	639.79 ± 57.67	621.43 ± 82.80	657.53 ± 43.81	774.28 ± 56.84	1159.37 ± 20.32	663.45 ± 53.62	1135.81 ± 163.75
5	Naringin	4293.99 ± 694.84	544.19 ± 57.61	2330.00 ± 62.10	43.33 ± 5.88	183.49 ± 21.41	493.23 ± 77.71	133.77 ± 22.64	1972.70 ± 308.36	1907.97 ± 298.14	155.73 ± 15.23	4980.87 ± 716.43	856.40 ± 27.96	1003.64 ± 121.72	2221.39 ± 43.38	966.65 ± 106.91	251.75 ± 39.35
6	Hesperidin	3874.78 ± 652.37	4160.72 ± 528.16	3091.37 ± 148.23	1195.08 ± 80.82	3060.41 ± 269.90	2626.72 ± 259.04	3575.26 ± 467.00	3227.13 ± 494.79	3102.85 ± 409.92	2480.56 ± 231.93	4758.26 ± 689.58	3177.08 ± 196.28	3868.27 ± 317.84	2890.70 ± 92.02	3181.91 ± 368.53	3125.96 ± 179.77
7	Hesperetin	5837.87 ± 264.68	3048.85 ± 303.57	2239.34 ± 96.43	814.26 ± 68.23	1993.62 ± 77.28	1807.70 ± 127.78	2145.84 ± 144.19	2330.46 ± 243.40	2295.72 ± 329.59	1834.12 ± 181.45	7156.59 ± 316.95	2489.32 ± 129.60	2786.83 ± 294.40	2006.73 ± 48.25	3308.82 ± 266.13	1839.25 ± 192.44
8	Scopoletin	150.36 ± 3.34	76.50 ± 12.71	69.38 ± 4.00	34.08 ± 5.08	64.44 ± 7.10	60.95 ± 1.04	44.94 ± 5.80	56.11 ± 6.83	86.36 ± 4.00	34.28 ± 4.00	237.55 ± 13.04	57.43 ± 2.46	64.86 ± 10.42	69.80 ± 4.72	69.86 ± 11.88	45.06 ± 3.03
9	Daidzein	14.06 ± 0.94	17.63 ± 1.57	9.12 ± 0.78	1.95 ± 0.28	9.00 ± 0.75	6.81 ± 0.91	6.31 ± 0.81	13.53 ± 1.14	12.57 ± 0.84	10.28 ± 1.19	6.89 ± 0.71	8.58 ± 1.20	9.99 ± 1.24	6.57 ± 0.83	9.20 ± 1.35	5.17 ± 0.10
10	Naringenin	2600.18 ± 217.74	379.36 ± 18.67	784.54 ± 27.63	35.69 ± 0.54	113.35 ± 12.30	206.63 ± 18.81	71.18 ± 4.94	425.68 ± 35.30	525.10 ± 70.04	227.03 ± 19.17	3259.16 ± 241.86	206.70 ± 8.61	246.75 ± 10.36	397.22 ± 10.40	153.62 ± 22.52	179.03 ± 12.65
11	Glycyrrhizic acid	3070.18 ± 175.29	4269.03 ± 199.04	3065.04 ± 77.75	1164.14 ± 183.71	3355.01 ± 426.61	2576.06 ± 255.23	4157.77 ± 211.33	3826.32 ± 335.32	3899.42 ± 542.40	3107.87 ± 378.10	2768.07 ± 250.76	2558.80 ± 180.03	2942.77 ± 268.86	3080.26 ± 131.21	3006.57 ± 333.22	3113.40 ± 136.04
12	Morin	3813.18 ± 383.83	708.37 ± 28.98	844.89 ± 111.80	105.58 ± 13.89	110.88 ± 16.48	563.48 ± 63.58	170.98 ± 23.36	539.10 ± 49.68	550.87 ± 49.07	311.26 ± 26.17	4019.50 ± 136.56	374.92 ± 20.03	449.90 ± 49.18	727.34 ± 59.61	379.26 ± 31.03	201.83 ± 22.11
13	Quercetin	2902.92 ± 420.22	464.34 ± 27.99	576.66 ± 67.05	81.35 ± 11.27	129.22 ± 16.14	455.71 ± 31.50	145.68 ± 12.20	378.52 ± 26.58	395.13 ± 41.98	252.14 ± 31.94	3107.47 ± 277.25	301.80 ± 24.18	344.56 ± 42.39	599.26 ± 51.07	304.66 ± 20.80	231.75 ± 22.44
14	Isoliquiritigenin	123.33 ± 1.54	107.88 ± 0.74	80.49 ± 2.98	17.05 ± 1.62	22.02 ± 1.40	42.85 ± 3.26	34.17 ± 5.03	142.57 ± 16.27	129.15 ± 11.10	65.46 ± 4.92	199.95 ± 8.58	16.57 ± 0.55	19.10 ± 0.64	61.64 ± 1.80	26.24 ± 1.75	17.41 ± 0.71
15	Liquiritigenin	392.14 ± 7.16	356.60 ± 0.22	263.01 ± 5.46	54.58 ± 5.33	67.60 ± 3.23	133.48 ± 17.50	113.34 ± 10.75	466.33 ± 60.63	408.01 ± 56.91	213.66 ± 28.54	649.33 ± 28.39	51.91 ± 2.90	58.18 ± 4.37	201.42 ± 3.37	85.16 ± 9.88	57.06 ± 6.40
16	Formononetin	146.95 ± 8.45	139.01 ± 3.45	107.96 ± 3.51	17.32 ± 2.04	110.97 ± 8.81	66.66 ± 9.18	55.63 ± 5.85	150.10 ± 11.65	92.85 ± 9.92	92.21 ± 7.64	115.78 ± 2.23	73.09 ± 3.82	81.79 ± 9.80	89.48 ± 4.13	74.02 ± 6.13	65.94 ± 5.70
17	Nobiletin	4129.53 ± 49.24	1783.50 ± 20.77	2515.31 ± 77.18	1850.06 ± 156.62	1954.99 ± 191.13	3735.75 ± 474.17	4389.54 ± 507.72	2561.99 ± 276.92	5054.44 ± 659.23	2330.66 ± 252.21	2747.98 ± 84.84	3001.93 ± 283.54	3325.35 ± 186.39	5093.37 ± 125.05	3912.02 ± 401.66	2206.13 ± 167.75
18	Tangeretin	2239.72 ± 63.67	692.66 ± 23.77	781.88 ± 35.83	764.14 ± 108.30	775.59 ± 80.63	1638.23 ± 199.89	1865.95 ± 205.54	750.53 ± 69.41	2278.55 ± 273.85	674.31 ± 71.45	1527.77 ± 102.02	1410.63 ± 114.69	1628.44 ± 71.47	2597.79 ± 53.12	1789.81 ± 206.46	684.47 ± 51.29
19	Obacunone	1826.45 ± 219.79	1392.87 ± 138.75	346.31 ± 12.17	178.50 ± 2.65	1142.22 ± 153.81	540.02 ± 80.23	725.34 ± 55.98	428.33 ± 51.34	303.89 ± 51.81	329.90 ± 10.00	1818.21 ± 207.43	481.09 ± 58.54	580.77 ± 17.44	224.79 ± 29.90	441.34 ± 26.66	298.26 ± 20.22
20	Glycyrrhetinic acid	40.62 ± 4.81	32.24 ± 1.36	19.98 ± 2.46	2.86 ± 0.20	14.83 ± 2.20	7.85 ± 0.78	11.66 ± 1.95	34.82 ± 4.88	31.69 ± 2.53	13.99 ± 0.90	80.30 ± 11.80	3.27 ± 0.21	4.79 ± 0.42	13.26 ± 0.93	3.68 ± 0.27	6.10 ± 0.46

WDD = Wendan Decoction.

**Table 3 molecules-28-01328-t003:** Regression data, LLOQs, precision, stabilities and repeatabilities for the 20 components of WDD.

NO.	Compound	Regression Equation	*r* ^2^	LLOQ ng/mL	Linear Range ng/mL	Precision RSD, *n* = 6	Stability RSD, 24 h, *n* = 6	Repeatability RSD, *n* = 6	Recovery % mean ± SD
1	Trigonelline	*Y* = 0.00299197*X* + 0.0378053	0.9904	120	120–60,000	13.98	7.44	8.54	94.48 ± 7.50
2	Synephrine	*Y* = 0.0900787*X* + 0.463625	0.9911	8	8–4000	13.79	6.91	11.87	104.74 ± 8.29
3	Isoquercitrin	*Y* = 0.00046952*X*−0.00753226	0.9907	140	140–70,000	14.75	13.05	10.49	94.13 ± 6.43
4	Liquiritin	*Y* = 0.00497658*X* − 0.00447113	0.9968	6	6–3000	8.83	3.63	11.53	93.91 ± 5.71
5	Naringin	*Y* = 0.00275605*X* − 0.0152202	0.9913	12	12–6000	11.31	6.63	12.38	102.78 ± 7.37
6	Hesperidin	*Y* = 0.00185764*X*−0.0162377	0.9926	24	24–12,000	10.96	5.71	13.37	100.97 ± 6.38
7	Hesperetin	*Y* = 0.00747729*X* + 0.0165338	0.9924	60	60–30,000	11.36	5.54	5.87	96.00 ± 5.06
8	Scopoletin	*Y* = 0.00145419*X*−0.0126843	0.9917	16	16–8000	12.2	7.17	7.38	101.27 ± 8.60
9	Daidzein	*Y* = 0.00364942*X* + 0.0178624	0.9948	4	4–2000	9.82	13.04	12.79	111.84 ± 10.65
10	Naringenin	*Y* = 0.0144335*X* − 0.0633108	0.993	14	14–7000	6.82	7.71	6.62	93.33 ± 5.30
11	Glycyrrhizic acid	*Y* = 0.0419993*X* − 0.197051	0.9905	10	10–5000	7.60	6.45	6.21	96.27 ± 9.18
12	Morin	*Y* = 0.0005182*X* + 0.130608	0.9932	20	20–10,000	10.6	13.25	11.03	97.35 ± 7.92
13	Quercetin	*Y* = 0.00131174*X*−0.00216878	0.9921	26	26–13,000	11.65	7.66	8.00	98.59 ± 10.04
14	Isoliquiritigenin	*Y* = 0.0370696*X* − 0.106264	0.9910	6	6–3000	8.00	7.50	4.77	111.43 ± 1.22
15	Liquiritigenin	*Y* = 0.0114821*X* + 0.0776073	0.9992	4	4–2000	10.33	6.54	8.89	89.62 ± 3.19
16	Formononetin	*Y* = 0.0197593*X* − 0.0374138	0.9903	4	4–2000	7.41	7.49	5.89	112.36 ± 3.13
17	Nobiletin	*Y* = 0.047895*X* − 0.0133269	0.9942	1	1–500	8.52	6.06	9.76	92.59 ± 3.64
18	Tangeretin	*Y* = 0.188519*X* + 0.0417842	0.9948	0.4	0.4–200	8.74	6.05	9.44	99.47 ± 8.74
19	Obacunone	*Y* = 0.000297133*X* − 0.00452012	0.9902	50	50–25,000	11.95	13.62	13.09	90.47 ± 7.57
20	Glycyrrhetinic acid	*Y* = 0.000755862*X* − 0.00133086	0.9946	60	60–30,000	11.89	6.39	12.95	102.81 ± 10.84

LLOQ = lower limit of quantification; WDD = Wendan Decoction.

**Table 4 molecules-28-01328-t004:** The inhibition rates of 16 batches of WDD (Mean ± SD, *n* = 3,).

Samples	0.25 mg/mL	0.5 mg/mL	1 mg/mL	2 mg/mL	4 mg/mL
S1	0.27 ± 0.16	0.38 ± 0.12	0.57 ± 0.05	0.24 ± 0.09	−0.03 ± 0.17
S2	0.19 ± 0.08	0.25 ± 0.10	0.53 ± 0.24	0.07 ± 0.02	−0.14 ± 0.12
S3	0.28 ± 0.20	0.33 ± 0.08	0.50 ± 0.13	0.10 ± 0.23	−0.10 ± 0.03
S4	0.19 ± 0.14	0.36 ± 0.07	0.59 ± 0.21	0.08 ± 0.23	−0.01 ± 0.16
S5	0.25 ± 0.25	0.32 ± 0.12	0.52 ± 0.16	0.08 ± 0.25	−0.10 ± 0.15
S6	0.18 ± 0.04	0.39 ± 0.26	0.58 ± 0.08	0.08 ± 0.06	−0.08 ± 0.21
S7	0.18 ± 0.04	0.39 ± 0.26	0.58 ± 0.08	0.08 ± 0.06	−0.08 ± 0.21
S8	0.18 ± 0.10	0.31 ± 0.11	0.61 ± 0.23	0.10 ± 0.18	−0.08 ± 0.25
S9	0.20 ± 0.15	0.30 ± 0.18	0.52 ± 0.13	0.17 ± 0.20	−0.22 ± 0.08
S10	0.12 ± 0.15	0.25 ± 0.11	0.48 ± 0.14	0.08 ± 0.11	−0.10 ± 0.11
S11	0.17 ± 0.19	0.26 ± 0.04	0.53 ± 0.12	0.22 ± 0.10	−0.05 ± 0.16
S12	0.12 ± 0.08	0.29 ± 0.14	0.47 ± 0.16	0.12 ± 0.10	−0.12 ± 0.18
S13	0.12 ± 0.10	0.26 ± 0.12	0.47 ± 0.06	0.06 ± 0.07	−0.11 ± 0.11
S14	0.09 ± 0.13	0.29 ± 0.21	0.47 ± 0.18	0.23 ± 0.05	0.04 ± 0.12
S15	0.20 ± 0.09	0.33 ± 0.15	0.52 ± 0.21	0.13 ± 0.15	−0.11 ± 0.13
S16	0.17 ± 0.13	0.33 ± 0.19	0.52 ± 0.26	0.16 ± 0.08	0.00 ± 0.13

WDD = Wendan Decoction.

**Table 5 molecules-28-01328-t005:** GRA and PLSR results for WDD compounds.

NO.	Compounds	GRA	PLSR	NO.	Compounds	GRA	PLSR
1	Trigonelline	0.88	0.822	11	Glycyrrhizic acid	0.901	−0.565
2	Synephrine	0.869	−0.507	12	Morin	0.795	0.856
3	Isoquercitrin	0.888	−0.193	13	Quercetin	0.798	1.044
4	Liquiritin	0.886	1.259	14	Isoliquiritigenin	0.828	0.349
5	Naringin	0.789	−0.513	15	Liquiritigenin	0.828	1.001
6	Hesperidin	0.892	0.404	16	Formononetin	0.883	1.477
7	Hesperetin	0.865	1.438	17	Nobiletin	0.864	−0.403
8	Scopoletin	0.843	0.521	18	Tangeretin	0.838	−0.716
9	Daidzein	0.885	−1.167	19	Obacunone	0.819	−1.003
10	Naringenin	0.785	−1.659	20	Glycyrrhetinic acid	0.829	−2.138

GRA = Gray relation analysis; PLSR = partial least squares regression; WDD = Wendan Decoction.

**Table 6 molecules-28-01328-t006:** Details of 16 batches of WDD.

Samples	BX	ZR	ZS	CP	GC
S1	Shandong	Guangdong	Jiangxi	Sichuan	Baotou, Inner Mongolia
S2	Shandong	Zhejiang	Chongqing	Fujian	Chifeng, Inner Mongolia
S3	Shandong	Hubei	Sichuan	Guangdong	Ordos, Inner Mongolia
S4	Shandong	Sichuan	Hunan	Hunan	Gansu
S5	Sichuan	Guangdong	Chongqing	Guangdong	Gansu
S6	Sichuan	Zhejiang	Jiangxi	Hunan	Ordos, Inner Mongolia
S7	Sichuan	Hubei	Hunan	Sichuan	Chifeng, Inner Mongolia
S8	Sichuan	Sichuan	Sichuan	Fujian	Baotou, Inner Mongolia
S9	Gansu	Guangdong	Sichuan	Hunan	Chifeng, Inner Mongolia
S10	Gansu	Zhejiang	Hunan	Guangdong	Baotou, Inner Mongolia
S11	Gansu	Hubei	Jiangxi	Fujian	Gansu
S12	Gansu	Sichuan	Chongqing	Sichuan	Ordos, Inner Mongolia
S13	Guangxi	Guangdong	Hunan	Fujian	Ordos, Inner Mongolia
S14	Guangxi	Zhejiang	Sichuan	Sichuan	Gansu
S15	Guangxi	Hubei	Chongqing	Hunan	Baotou, Inner Mongolia
S16	Guangxi	Sichuan	Jiangxi	Guangdong	Chifeng, Inner Mongolia

WDD = Wendan decoction; BX = Breit; ZR = Stapf ex Rendle; ZS = Citrus aurantium L.; CP = Citrus reticulata Blanco; GC = Glycyrrhiza uralensis Fisch.

## Data Availability

The datasets during and/or analysed during the current study available from the corresponding author on reasonable request.

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
