# Peer review of "Screening of Active Ingredients from Wendan Decoction in Alleviating Palmitic Acid-Induced Endothelial Cell Injury"

_molecules, 2023, doi:10.3390/molecules28031328_

Round 1
Reviewer 1 Report
Please refer to the attachment!

Author Response
Point-by-point response to reviewers
Dear Editors,
Thank you for giving us the opportunity to revise our manuscript entitled “Screening of active ingredients from Wendan Decoction in alleviating palmitic acid-induced endothelial cell injury". We appreciate reviewers for the positive and constructive comments and suggestions on our manuscript. You will find our detailed response in the following pages. The major changes in the revised manuscript have been highlighted in red.
Thank you again for considering our revised manuscript for publication. Please do not hesitate to contact us in case of any question.
In the manuscript " Screening of active ingredients from Wendan Decoction in alleviating palmitic acid-induced endothelial cell injury", the author investigated the protective effects of WDD on endothelial cells using 16 batches of WDD tests. The paper fits the aims and scope of Molecules. The paper seems to be acceptable but it requires some modifications.
- Line 105: The chemical composition of WDD was obtained based on previous study (Hong, Fang et al.2021). This reference literature conducted an investigation of Gegen Decoction. However, can the authors decide that the primarily active constituents are the same in two prescriptions?
Authors’ reply: We are thankful to the reviewers for pointing it out. We are regretful for the misunderstanding caused by our incorrect expression. In fact, we identified the ingredients in WDD by referring to the methods in the literature. In fact, the ingredients in the two prescriptions are different. We have modified this sentence as follows:
Line 101: The identification method of chemical constituents in WDD refer to previous study (Hong, Fang et al.2021).
- Line 147: The above operations were repeated to obtain WDD solution and then concentrated with a rotary evaporator. Please provide the specific conditions used for the rotary evaporation.
Authors’ reply: We’re thankful for this precious comment. Our pleasure to add
the required details.
Line 137: The above operations were repeated to obtain WDD solution and then
concentrated with a rotary evaporator(Temperature: 60 °C; Rotational speed: 40~
80rpm).
- Line 159,181: All standard solutions were weighed accurately according to international standards...; ...with internal standard method. However, what are the international standards (method)?
Authors’ reply: Thanks very much for your valuable comment. We are very sorry for the faulty expression. We have made the following modifications:
Line 147: All standard solutions were weighed and diluted in methanol to form a reserve solution, which was stored at 4 ℃.
- Line 62: so the clinical evaluation of endothelial cells may be… In the scientific literature there is a certain rigor which does not allow “new star”or “so” like expressions. “So” synonyms = for that reason, as a result, accordingly, therefore, then, thus, consequently etc.
Authors’ reply: We extend gratitude to reviewers on this guidance. We have made the following modifications:
Line 63: Experimental and clinical research has indisputably associated endothelium with the pathogenesis of hyperlipidemia. Therefore, the clinical evaluation of endothelial cells may be considered to evaluate vascular wellbeing (or illness) (Incalza, Maria Angela et al. 2018).
- Line 176: The gradient elution (is)as follows:…;Line 176: and (the)repeated injection analysis…;Line 209: ,(and)each batch of concentrated arrangement was…. The English language must still be revised.
Authors’ reply: Thank You very much for giving us the guideline. We have made the following changes:
Line 160: The gradient elution conditions of mobile phase B are as follows: 0–0.5 min: 5%, 0.5–2 min: 5%–8%, 2-2.1 min: 40%, 2.1–4 min: 40%-50%, 4–6 min: 50%–60%, 6–6.1 min: 60%–70%, 6.1–8 min: 70%-80%, 8–8.1 min: 80%-5%, 8.1–10 min: 5%. The sample size for analysis was 3 μL.
Line 168: The sample solution was put into the sample bottle at room temperature and injected for analysis at different times (0, 4, 8, 12, 16, 24h) after preparation to observe the sample stability at room temperature.
Line 187: The 16 batches of WDD powder were absolutely weighed. The volume was settled to 1 mL by PBS, and the microscopic organisms were sifted by 0.22 μm channel layer, later the cells were included, and each batch of concentrated arrangement was weakened to 5 concentrations of 0.25 mg/mL, 0.5 mg/mL, 1 mg/mL, 2 mg/mL, 4 mg/mL by RPMI-1640 basal medium with 10% FBS.
- Line 218-222:Concurring to the examination results of each component of WDD,… Please rewrite this sentence.
Authors’ reply: We’re pleased to make the modifications as follows:
Line 196: The data reflecting the pharmacodynamic behavior of WDD system was taken as the reference series, and the quantitative data of each component that might affect the pharmacodynamic behavior of WDD system was taken as the comparison series.
- Line 238:..and IS were shown in Figure 1. You should present the full name where an acronym is in the first use.
Authors’ reply: Thank you so much for pointing this out. We have changed it to the full name as follows:
Line121: Obacunone (111923-201102) and furosemide (internal standard, IS) (100544-201503) were obtained from National Institutes for Food and Drug Control.
- Line 252:.All the standard twists were built up concurring to… Please rewrite this sentence.
Authors’ reply: We thank you for this guidance. We have rewritten it as follows:
Line 224: The slopes, intercepts and correlation coefficients (r2) were obtained by weighted (1/x) linear regression analysis.
- The format of the references is confused. Please modify the format of the reference 5,38,42 and 43.
Authors’ reply: We are grateful for this guideline. We have made the changes as follows:
- is replaced by 32.Wang, Q., Zou, Z., Zhang, Y., Lin, P., Lan, T., Qin, Z., Xu, D., Wu, H., Yao, Z. (2021). Characterization of chemical profile and quantification of major representative components of Wendan decoction, a classical traditional Chinese medicine formula. Journal of separation science,44(5):1036-1061. https://doi.org/10.1002/jssc.202000952
38.is replaced by 16.Li,T., Zhang,L., Jin,C., Xiong,Y., Cheng,Y,Y., Chen,K.(2020). Pomegranate flower extract bidirectionally regulates the proliferation, differentiation and apoptosis of 3T3-L1 cells through regulation of PPARγ expression mediated by PI3K-AKT signaling pathway. Biomedicine & pharmacotherapy,131,110769. https://doi.org/10.1016/j.biopha.2020.110769
- is replaced by 4.Chen,M, Yang,F, Yang,X, Lai,X, Gao,Y.(2016) Systematic Understanding of Mechanisms of a Chinese Herbal Formula in Treatment of Metabolic Syndrome by an Integrated Pharmacology Approach. International journal of molecular sciences,17(12):2114. https://doi.org/10.3390/ijms17122114
43.is deleted.

Reviewer 2 Report
The authors identified 20 ingredients from WDD which have shown apparent protective effects on HUVEC cells. GRA and PLSR analysis indicated trigonelline, liquiritin, hesperidin, hesperetin, scopoletin, morin, quercetin, isoliquiritigenin, liquiritigenin and formononetin may be active components. Overall, this article performed an interesting study. But I have the following concerns.
(1) GRA and PLSR analysis provided 10 active components, whether these compounds have protective effects?
(2) The experimental results showed that cell proliferation was dose-dependent with PA concentration, why 200 μM was selected rather than 150 μM or 300 μM?
(3) Figure 2 and Figure 3 are not consistent with the text, please check. And Figure 3 are not qualified for publication.
(4) Some Tables like Table 2-5 should be removed to SI to make the manuscript clearer. And I think it would be better to introduce the protective effect of WDD prior to the identification.
Author Response
The authors identified 20 ingredients from WDD which have shown apparent protective effects on HUVEC cells. GRA and PLSR analysis is indicated trigonelline, Liquiritin, hesperidin, hesperetin, scopoletin, morin, quercetin, isoliquiritigenin, liquiritigenin and formononetin may be active components. Overall, this article performed an interesting study. But I have the following concerns.
- GRA and PLSR analysis provided 10 active components, whether these compounds have protective effects?
Authors’ reply: We are pleased to explain here. These ten components were identified by statistical analysis as substances that may have cell protective effects in WDD. Combined with literature studies, we also discussed the pharmacologic effects related to components in the discussion, but the activity of monomer components was not studied in this study.
- The experimental results showed that cell proliferation was dose-dependent with PA concentration, why 200 uM was selected rather than 150 uM or 300 uM?
Authors’ reply: Thanks for this valuable comment. Actually, We chose LPS concentration of IC50 around 50% as our molding concentration. Therefore, we chose 200 uM.
- Figure 2 and Figure 3 are not consistent with the text. please check. And Figure 3 are not qualified for publication.
Authors’ reply: We extend gratitude to reviewers on this guidance. We have checked the contents of Figure 2 and 3 with the text and improved the resolution of Figure 2 and 3. At the same time, we found that the compound number 10 in Figure 1 was labeled 9, which we have corrected.
- Some Tables like Table 2-5 should be removed to Sl to make the manuscript clearer. And l think it would be better to introduce the protective effect of WDD prior to the identification.
Authors’ reply: Thank you for your kind suggestion. The modification suggestions put forward by experts are much valuable. It is indeed easier to read and understand if the drug effect is described first and then component identification is carried out. However, the modification will cause significant changes in the article, but it will not affect the design idea and content of the study. Therefore, we ask the editor and the review experts to consider again. If necessary, we can adjust the internal order.

Round 2
